# Association of Maternal Anemia and Cesarean Delivery: A Systematic Review and Meta-Analysis

**DOI:** 10.3390/jcm12020490

**Published:** 2023-01-06

**Authors:** Ishag Adam, Yasir Salih, Hamdan Z. Hamdan

**Affiliations:** 1Department of Obstetrics and Gynecology, Unaizah College of Medicine and Medical Sciences, Qassim University, Unaizah 51911, Saudi Arabia; 2Faculty of Medicine, University of Khartoum, P.O. Box 102, Khartoum 11111, Sudan; 3Department of Basic Medical Sciences, Unaizah College of Medicine and Medical Sciences, Qassim University, Unaizah 51911, Saudi Arabia; 4Department of Biochemistry and Molecular Biology, Faculty of Medicine, Al-Neelain University, Khartoum 11121, Sudan

**Keywords:** anemia, cesarean delivery, pregnancy, meta-analysis

## Abstract

Anaemia during pregnancy is associated with an increased incidence of caesarean delivery (CD). This study was conducted to explore the association between CD and maternal anaemia. The PubMed/MEDLINE, Cochrane, Google, Google Scholar and ScienceDirect databases were searched for relevant studies on this topic. The assessment and review were conducted with the Joanna Briggs Institute Meta-Analysis of Statistics Assessment and Review Instrument. The studies were assessed using the modified Newcastle–Ottawa quality assessment scale. Data were collected in an Excel sheet, and the ‘meta’ package of the R 4.0.3 software was used for statistical analysis. Fourteen studies that enrolled 336,128 pregnant women were included in the meta-analysis. Women with anaemia were found to be at a higher risk for CD (OR = 1.63, 95% CI = 1.23–2.17). As heterogeneity was detected in the studies, the random-effects model was used for the pooled meta-analysis (Q = 96.7, *p* < 0.001). In the subgroup analysis, anaemic women were found to be at higher risk for CD in studies from both low-middle-income (7) and high-income countries (7). In meta-regression analysis, none of the investigated covariates were associated with the pooled OR of CD. This evidence demonstrates with a moderate level of certainty that anaemic pregnant women are more likely to have CD than non-anaemic pregnant women.

## 1. Introduction

Pregnant women (especially in developing countries) are more vulnerable to anaemia than pregnant women from developed countries, and anaemia has been found to lead to adverse maternal and perinatal outcomes [1]. The World Health Organization (WHO) defines anaemia during pregnancy as a haemoglobin level of less than 11 g/dL, while the Centers for Disease Control and Preventions (CDC) and American College of Obstetricians and Gynecologists (ACOG) consider a haemoglobin level of less than 10.5 g/dL during the second trimester as the criterion for diagnosing anaemia [2,3,4]. The WHO considers anaemia to be a serious public health problem when its prevalence is equal to or more than 40% [5]. It has been estimated that, globally, anaemia affects 56% of pregnant women in low- and middle-income countries [1]. Further, the prevalence of anaemia in pregnant women is 57% in sub-Saharan Africa and 48% in South-East Asia, while it is much lower in South America at 24.1% [5].

Anaemia in pregnant women has been reported to have various adverse effects on both the mother and unborn foetus. These effects include preterm birth, a low birth weight and an increase in the risk of maternal and perinatal mortality [6,7]. In particular, recent studies have shown that anaemia is associated with an increased risk of caesarean delivery (CD) [8,9,10]. While certain adverse maternal and perinatal effects of anaemia, such as low birth weight, are well-covered in the literature and even in meta-analyses [11], to our knowledge, no systemic review and meta-analysis has been conducted on the association between anaemia and CD. Investigating the predictors of and factors associated with CD is important because the occurrence of CD has been increasing in both developed and developing countries [12]. Therefore, this systematic review and meta-analysis was conducted to investigate the association between maternal anaemia and CD.

## 2. Methods

### 2.1. Study Design and Search Strategy

The PRISMA guidelines for reviews and meta-analyses were strictly followed [13] (Appendix A). The initial search strategy involved the use of free text words, words in titles or abstracts, and MeSH terms for each exposure condition, group of participants, and study design separately. The terms used in these searches were combined using Boolean operators. Eligible studies on anaemia and CD were further searched by reviewing the reference lists of the identified articles. The major databases PubMed/MEDLINE, Cochrane library, Google Scholar and ScienceDirect were used to identify articles. The key terms and search strategy used in the databases are presented in detail in Appendix A.

### 2.2. Inclusion Criteria

All studies on the association between CD and maternal anaemia were included in the current systematic review and meta-analysis. Maternal anaemia is defined as haemoglobin level below 10.5 g/dL according to the CDC recommendation.

### 2.3. Study Design

We included cross-sectional, case-control and cohort studies.

### 2.4. Language

Only studies published in the English language were included.

### 2.5. Population

Studies conducted in all regions of the world that investigated CD and anaemia were considered.

### 2.6. Publication and Publication Year

Articles published until October 2022 were included, and no starting date restrictions were applied.

### 2.7. Exclusion Criteria

Articles published in languages other than English, case reports, review articles, studies on congenital anaemia and studies which did not report specific outcomes for CD and anaemia were excluded.

### 2.8. Quality Assessment and Assessment of Risk of Bias

The quality of each study was assessed using the Newcastle–Ottawa Scale (NOS) for quality assessment of cohorts and case-control studies, and the modified NOS for cross-sectional studies [14]. A maximum of nine stars were awarded to each study based on three major domains: selection of participants, comparability of study groups and ascertainment of outcomes of interest. Studies that were assigned ≥7 stars were considered as high-quality studies. Two investigators (YS and HZH) independently identified eligible studies for this meta-analysis. Any disagreement was resolved by discussion with the other researcher (IA). The Cochrane collaboration tool was used to assess each included study for possible risk of bias.

### 2.9. Data Extraction

The Joanna Briggs Institute Meta-Analysis of Statistics Assessment and Review Instrument (JBI-MAStARI) was used to extract data from the included studies [15]. For each included study, the following data were extracted in an Excel sheet: author’s name, year of publication, country represented by the population, number of anaemic women, number of non-anaemic women, number of women with CD in the anaemia and non-anaemic group, haemoglobin level, and the trimester in which the haemoglobin level was investigated.

### 2.10. Statistical Analysis

The meta-analysis was conducted using the ‘metabin’ function of the ‘meta’ package [16] of the R 4.03 software (The R Foundation for Statistical Computing, Vienna, Austria). The heterogeneity of the included studies was evaluated based on Cochrane Q and the I^2^ statistic, which were calculated with the χ^2^-based Q test and *I^2^* test, respectively. Cochrane Q values with *p*  <  0.10 and I^2^  >  50 were considered to indicate the presence of heterogeneity between the included studies [17]. Sensitivity analysis was performed along with the Baujat plot to identify studies that contributed to heterogeneity and significantly affected the pooled estimates. The fixed-effects model swas used for analysis when there was no evidence for significant heterogeneity between studies (*p* > 0.10, *I^2^* < 50%), while the random-effects model was used for studies with significant heterogeneity. Planned subgroup analysis of the studies was performed based on the World Bank economic classification system of sorting countries into low-income and high-income countries and the cut-off point for haemoglobin level (>10.5 g/dL vs. <10.5 g/dL). Meta-regression analysis was performed to identify which covariates (year of publication, NOS quality score, country income level, study continent, haemoglobin cut-off level and trimester of haemoglobin measurement) significantly affect the OR of CD. Egger’s test, in addition to the funnel plot, was used to assess for publication bias. *P*-values were considered to indicate statistical significance if they were <0.05.

### 2.11. Assessment of the Certainty Level of Evidence

The level of certainty for the generated evidence was determined with the online open access tool GRADEpro GDT [18]. This software tool relies on the evidence for the risk of publication bias, inconsistency of results, indirectness of evidence and imprecision.

## 3. Results

### 3.1. Study Selection

The selection and characteristics of the included studies are shown in Figure 1. With the initial search algorithm, 146 studies were retrieved. After the elimination of some studies based on the exclusion criteria, the remaining studies (n = 23) were further reviewed by reading the full text. An additional nine studies were then excluded because they did not provide specific data about CD or anaemia. Therefore, 14 studies conducted between 2002 and 2019 that enrolled a total 336,128 pregnant women were finally included in our meta-analysis [6,8,9,10,19,20,21,22,23,24,25,26,27]. All 14 studies were good quality studies and had the least risk of bias according to the NOS score (Table 1) and Cochrane collaborate tool (Figure 2).

### 3.2. Features of the Included Studies

Of the fourteen studies, two were conducted in Europe [21,26], three in Africa [22,23,27], three in India [9,19,24], two in Israel [6,10], one in New Zealand [8], one in Jordan [25], one in Pakistan [28], and one in Iraq [20]. The number of included women ranged from 382 [8] to 153,396 [6]. Nine studies used the WHO cut-off of 11 g/dL haemoglobin for the diagnosis of anaemia [8,9,10,19,20,21,22,23,25,26,27], while five studies adopted the CDC cut-off of <10.5 g/dL [6,22,23,24,28] (Appendix A). In over half of the studies (n = 9, 64.2%) [10,19,20,21,22,23,24,25,28], haemoglobin was measured in the third trimester or at delivery, while it was measured in the first trimester in four studies [6,8,9,24] and the second trimester [26] in one study.

### 3.3. Association of CD with Maternal Anemia

A total of 336,128 pregnant women were enrolled in the 14 included studies. Of the 336,128 included women, 93,046 (27.6%) were diagnosed with anaemia. The incidence of CD was significantly higher in pregnant women with anaemia than in pregnant women without anaemia (34,324/93,046 [36.8%] vs. 25,956/243,082 [10.6%], *p* < 0.001). Overall, women with anaemia were at a higher risk for CD (OR = 1.63, 95% CI = 1.23–2.17) (Figure 3). Because heterogeneity was detected between the reviewed studies, the random-effects model was used for the meta-analysis (Q = 96.0, *p* < 0.01). The Baujat plot revealed that the Levy et al. (2000) study contributed significantly to heterogeneity and influenced the overall estimate [6] (Figure 4). However, sensitivity analysis showed that it was non-significant, and, therefore, we did not remove it from the random-effects model (Figure 5).

### 3.4. Subgroup and Meta-Regression Analysis

In the subgroup analysis, anaemic women were found to be at higher risk for CD in studies from both low-middle-income (n = 8) [8,20,21,22,25,26,27,28] and high-income countries (n = 6) [6,8,10,21,23,26] (Figure 6). Moreover, anaemic women were at higher risk for CD regardless of the haemoglobin cut-off point (10.5 g/dL or 11 g/dL) (Figure 7).

Meta-regression analysis of potential covariates that may affect the pooled OR of CD showed that none of the investigated covariates significantly affected the pooled OR (Table 2).

### 3.5. Assessment of Publication Bias

A funnel plot showed that there was no evidence of visual asymmetry of the plotted studies (Figure 8), and this was confirmed by Egger’s test (*p* = 0.711).

### 3.6. Level of Evidence

According to the GRADEpro tool, the current generated evidence was of moderate certainty for the measured outcome, which was the OR of CD among anaemic pregnant women (Table 3).

## 4. Discussion

In the current review and meta-analysis, the risk of CD was 1.63 times higher for anaemic pregnant women than for non-anaemic pregnant women. While various maternal and perinatal outcomes of anaemia in pregnancy, such as low birth weight and preterm delivery, have been well investigated, the association between anaemia and CD has been investigated to a much lesser extent [9,10,11]. Furthermore, anaemia has not been included in traditional pregnancy outcome indicators, such as age, parity, education and antenatal care for CD [29]. With the exception of one study [27], the remaining studies investigated the general maternal and perinatal outcomes, including CD. The reasons for the higher rate of CD among anaemic pregnant women are not yet fully understood. One of the reasons could be foetal distress, as maternal anaemia has been found to cause foetal distress [30,31]. Further, anaemia can impair the delivery of oxygen to tissues, including the uterus, thereby causing uterine inertia during labour [32], and it may also reduce oxygen delivery to the uterus (and the foetus) [33]. Poor placentation and placental insufficiency, which are features of the placenta in pregnant women who are anaemic, might also explain the increased rate of CD among anaemic women [34,35]. Recently, it was found that foetal circulation in anaemic mothers is characterised by a significant reduction in the resistance index of the middle cerebral artery and the resistance ratio of the middle cerebral/umbilical artery [36]. Similarly, anaemia might alter blood viscosity and, hence, blood flow through the placental and inter-villous space [37].

Maternal anaemia in the third trimester has been reported to be associated with foeto-maternal complications such as low birth weight and preterm birth [38]. Based on these findings, we hypothesised that anaemia in the third trimester may result in a higher risk of CD than anaemia in the first trimester. This is probably because detection of anaemia in the first trimester may lead to its treatment before the time of delivery. However, detection in the first trimester is difficult in sub-Saharan countries, such as Sudan and South Africa, as pregnant women in these countries report late for antenatal care. In most of the included studies in this review (n = 9, 64.2%) [10,19,20,21,22,23,24,25,28], haemoglobin was measured during the third trimester, while four studies [6,8,9,24] measured it in the first trimester and one study measured it in the second trimester [26]. We investigated the impact of the time at which haemoglobin was measured on the risk of CD, but it was not significant. This may indicate that anaemic pregnant women are at risk of CD regardless of the time of onset of the anaemia. 

The current meta-analysis has several limitations. The first limitation is the use of different cut-off values for haemoglobin for the diagnosis of anaemia, as this may have led to the significantly high heterogeneity observed between the studies. The second limitation is the different time of haemoglobin measurement during pregnancy in the included studies, as this may also have contributed to the high heterogeneity observed. Third, data on the presence of haemoglobinopathies were not reported in the included studies; therefore, the potential impact of this group of disorders could not be assessed. Finally, differences in the ethnicities of the included populations and indicators of anaemia could also have contributed to the high heterogeneity of the studies and need to be considered when interpreting the findings.

## 5. Conclusions

The findings of this systematic review and meta-analysis indicate that anaemic pregnant women are more likely to have CD than non-anaemic pregnant women, regardless of the time of onset of the anaemia. In the future, more studies are needed to assess the effects of anaemia on the incidence of CD and to investigate the mechanism underlying the association between anaemia and CD.

## Figures and Tables

**Figure 1 jcm-12-00490-f001:**
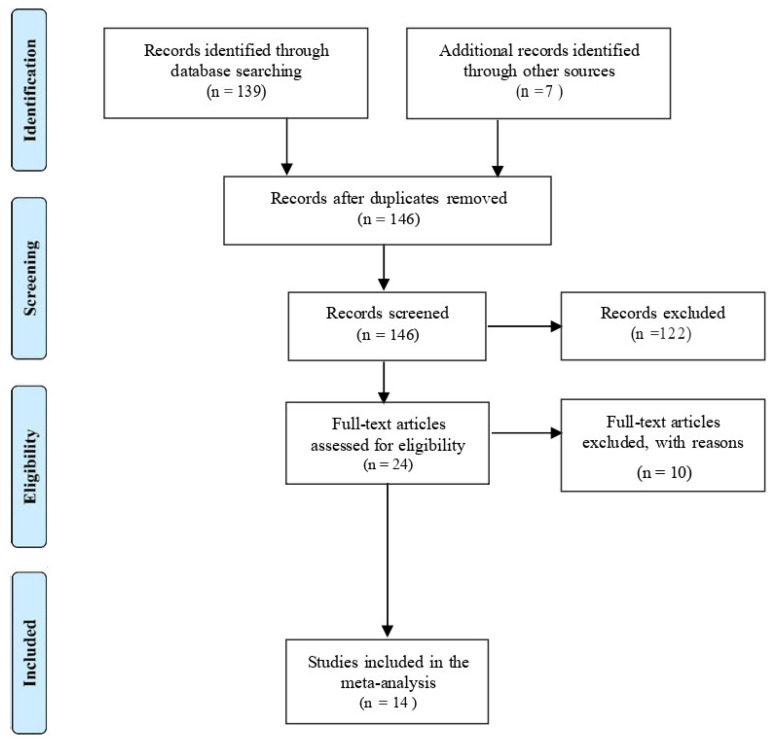
Flow diagram showing the number of articles identified in the systematic review and meta-analysis of maternal anaemia and caesarean delivery.

**Figure 2 jcm-12-00490-f002:**
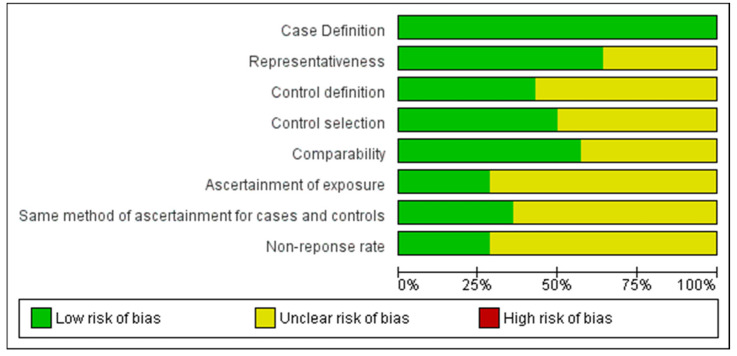
Risk of bias assessment graph and summary.

**Figure 3 jcm-12-00490-f003:**
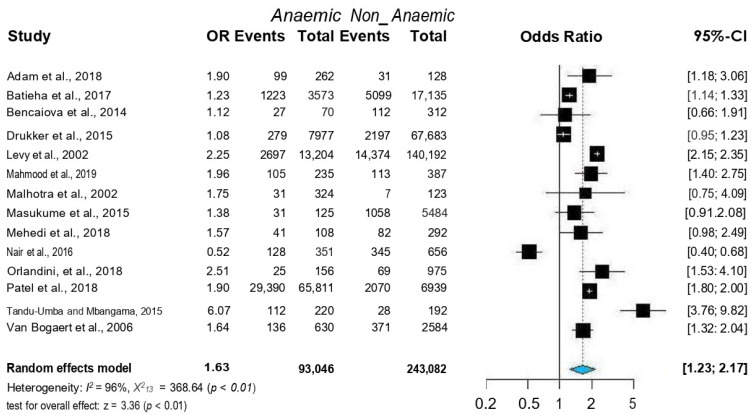
Forest plots of ORs with 95% CIs of the meta-analysis of anaemia and caesarean delivery. Published data from Adam et al. [27], Batieha et al. [25], Bencaiova et al. [26], Drukker et al. [10], Levy et al. [6], Mahmood et al. [28], Malhotra et al. [19] Masukume et al. [8] Mehedi et al. [20] Nair et al. [24] Patel et al. [9] Orlandini et al. [21] Tandu-Umba et al. [22] and Van Bogaert et al. [23] were used to calculate the pooled OR.

**Figure 4 jcm-12-00490-f004:**
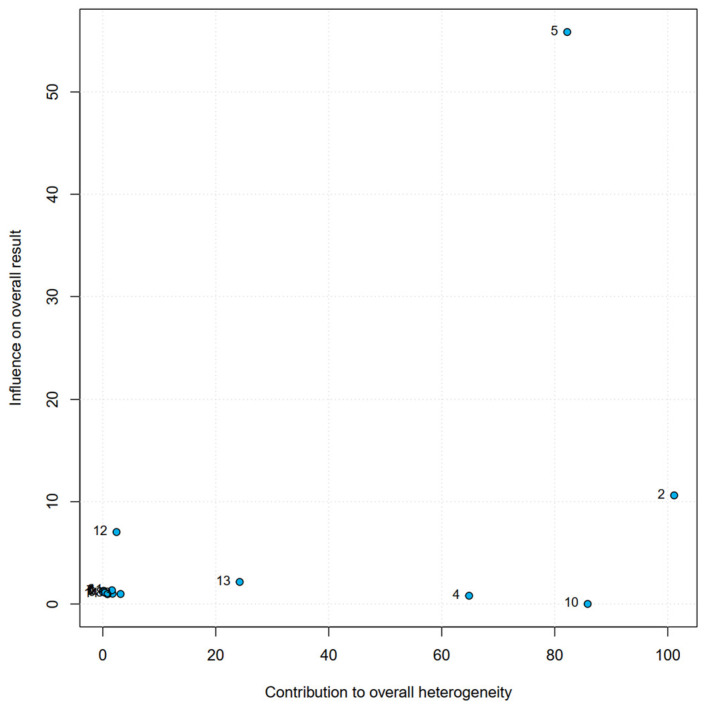
Baujat plot. Published data from 1 (Adam et al. [27]), 2 (Batieha et al. [25]), 3 (Bencaiova et al. [26]), 4 (Drukker et al. [10]), 5 (Levy et al. [6]), 6 (Mahmood et al. [28]), 7 (Malhotra et al. [19]), 8 (Masukume et al. [8]), 9 (Mehedi et al. [20]), 10 (Nair et al. [24]), 11 (Patel et al. [9]), 12 (Orlandini et al. [21]), 13 (Tandu-Umba et al. [22]) and 14 (Van Bogaert et al. [23]) were used to calculate the pooled OR and heterogeneity.

**Figure 5 jcm-12-00490-f005:**
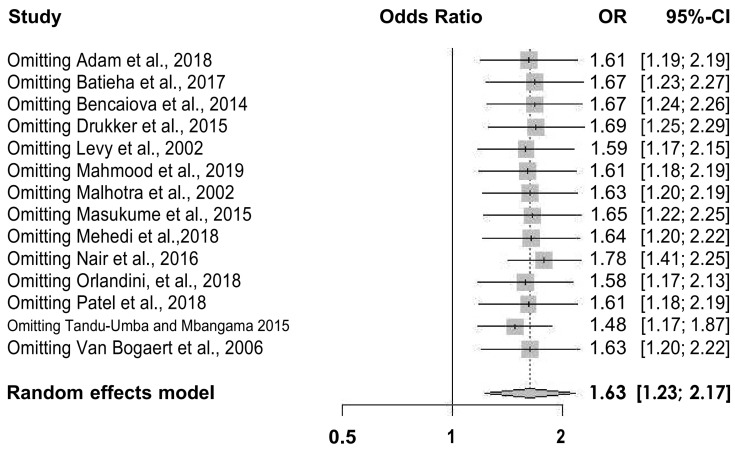
Sensitivity analysis performed by removing one study at a time. Published data from Adam et al. [27], Batieha et al. [25], Bencaiova et al. [26], Drukker et al. [10], Levy et al. [6], Mahmood et al. [28], Malhotra et al. [19] Masukume et al. [8] Mehedi et al. [20] Nair et al. [24] Patel et al. [9] Orlandini et al. [21] Tandu-Umba et al. [22] and Van Bogaert et al. [23] were used to calculate the pooled OR and perform the sensitivity analysis.

**Figure 6 jcm-12-00490-f006:**
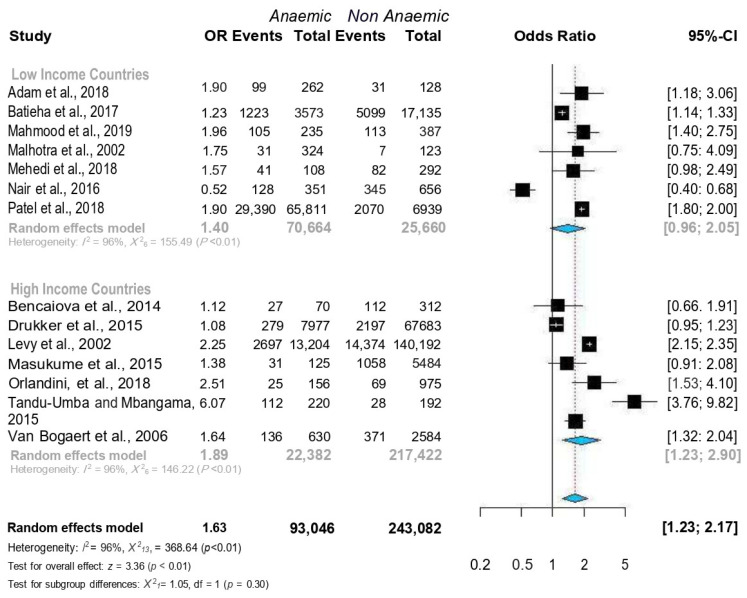
Forest plots of ORs with 95% CIs of the meta-analysis of the anaemia and caesarean delivery subgroup analysis according to World bank classifications. Published data from Adam et al. [27], Batieha et al. [25], Bencaiova et al. [26], Drukker et al. [10], Levy et al. [6], Mahmood et al. [28], Malhotra et al. [19] Masukume et al. [8] Mehedi et al. [20] Nair et al. [24] Patel et al. [9] Orlandini et al. [21] Tandu-Umba et al. [22] and Van Bogaert et al. [23] were used to calculate the pooled OR.

**Figure 7 jcm-12-00490-f007:**
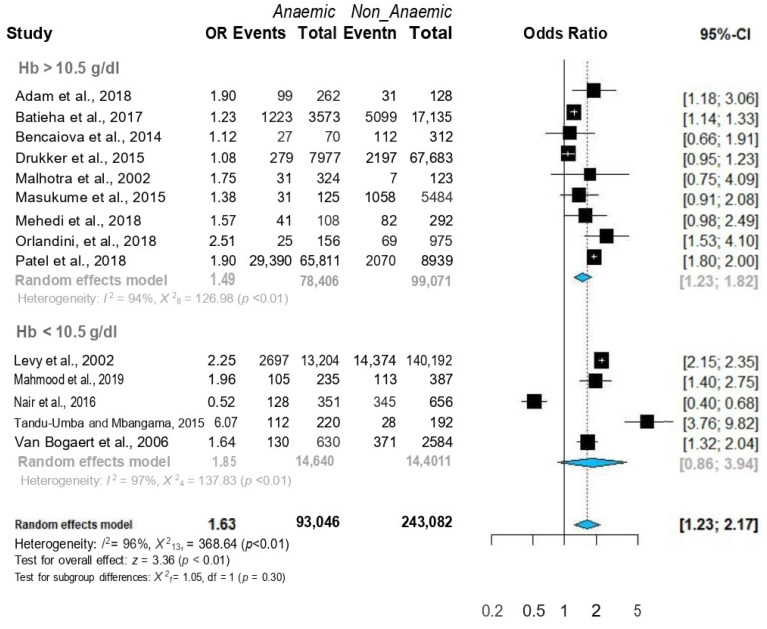
Forest plots of ORs with 95% CIs of the meta-analysis of the anaemia and caesarean delivery subgroup analysis according to Hb cut-off value. Published data from Adam et al. [27], Batieha et al. [25], Bencaiova et al. [26], Drukker et al. [10], Levy et al. [6], Mahmood et al. [28], Malhotra et al. [19] Masukume et al. [8] Mehedi et al. [20] Nair et al. [24] Patel et al. [9] Orlandini et al. [21] Tandu-Umba et al. [22] and Van Bogaert et al. [23] were used to calculate the pooled OR.

**Figure 8 jcm-12-00490-f008:**
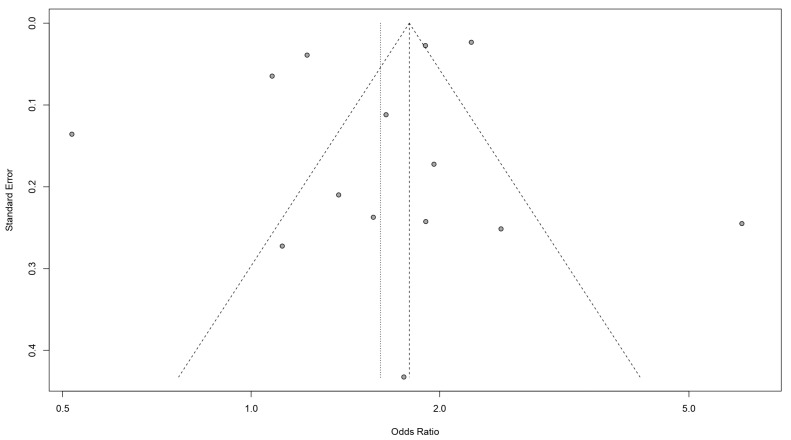
Funnel plot for the publication bias.

**Table 1 jcm-12-00490-t001:** Newcastle-Ottawa Scale rating for included studies: (* indicates the number of the star for each criterion, Maximum score = 9 stars).

Study	Selection	Comparability	Outcome	Total Score
Adam et al. [27]	****	**	***	9
Batieha et al. [25]	****	**	**	8
Bencaiova et al. [26]	****	**	**	8
Drukker et al. [10]	****	**	***	9
Levy et al. [6]	****	**	**	8
Mahmood et al. [28]	****	**	**	8
Malhotra et al. [19]	****	**	**	8
Masukume et al. [8]	****	***	*	8
Mehedi et al. [20]	****	**	**	8
Nair et al. [24]	****	**	**	8
Patel et al. [9]	****	**	**	8
Orlandini et al. [21]	****	***	*	8
Tandu-Umba et al. [22]	****	**	**	8
Van Bogaert et al. [23]	****	**	**	8

**Table 2 jcm-12-00490-t002:** Meta-regression analysis of the factors associated with anaemia and caesarean delivery.

Covariate	Coefficient	95% Confidence Interval	Standard Error	*p*
**Continents** Asia Europe	−0.847 −1.710	(−1.899, 0.204) (−3.810, 0.389)	0.536 1.071	0.114 0.110
**Haemoglobin cut-off level** Hb > 10.5 g/dl	0.864	(−0.470, 2.199)	0.681	0.204
**Income levels** Low-income countries	−0.842	(−2.003, 0.317)	0.592	0.154
Year of publication	0.009	(−0.059, 0.077)	0.035	0.797
Study NOS quality score	−1.002	(−2.162, 0.158)	0.592	0.090
**Hemoglobin measurement time** Third trimester	−0.004	(−0.833, 0.824)	0.423	0.991

**Table 3 jcm-12-00490-t003:** GRADE table. Association of CD with maternal anemia compared.

Certainty Assessment							№ of Patients		Effect		Certainty	Importance
№ of Studies	Study Design	Risk of Bias	Inconsistency	Indirectness	Imprecision	Other Considerations	Table 3. GRADE Table. Association of CD with Maternal Anemia	Not Anemia	Relative (95% CI)	Absolute (95% CI)		
Association of CD with maternal anemia												
14	observational studies	not serious	not serious	not serious	not serious	all plausible residual confounding would reduce the demonstrated effect	93046 cases 243082 controls		OR 1.63 (1.23 to 2.17)	-	⨁⨁⨁◯ Moderate	IMPORTANT
							-	3.4%		2 more per 100 (from 1 more to 4 more)		
							-	24.8%		10 more per 100 (from 4 more to 17 more)		
							-	50.9%		12 more per 100 (from 5 more to 18 more)		

CI: confidence interval; OR: odds ratio.

## Data Availability

The datasets used and/or analysed during the current study are available within the paper.

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
