# Peer review of "Association of Maternal Anemia and Cesarean Delivery: A Systematic Review and Meta-Analysis"

_jcm, 2023, doi:10.3390/jcm12020490_

Round 1

Reviewer 1 Report

The authors performed a systematic review to examine the effect of maternal anemia on the rate of cesarean delivery. This study is of some interest; however, this study has severe methodological flaws that can not be addressed. The reviewer’s comments are as follows.

Recently, a similar systematic review has been published (Obstet Gynecol Surv. 2022 Oct;77(10):595-605.).

Among the eligible studies, following points were varied among the studies.

1.    The timing of blood test (Hb assessment)

2.    The definition of anemia

3.    Inclusion or exclusion of maternal blood disorder (i.e. thalassemia).

The authors included planned cesarean delivery such as repeat cesarean delivery.

The authors did not consider the etiology of maternal anemia.

Overall, due to the severe heterogeneity among the studies the effect of anemia on the rate of cesarean delivery is difficult to assess.

Author Response

Reviewer 1

Comment

  1. Recently, a similar systematic review has been published (Obstet Gynecol Surv. 2022 Oct;77(10):595-605.).

Response

The authors much appreciate your positive and helping feedback. Yes, this is systematic review without meta-analysis. Our is both systematic review and meta-analysis and as you know the meta-analysis is the statistical part (core) of the process. Now (using our results), it can be concluded that anemic women are at 1.63 at higher risk to have CD (OR=1.63) and of course this can not be stated using the previous study.

Comment

  1. Among the eligible studies, following points were varied among the studies.

1.The timing of blood test (Hb assessment)

2.The definition of anemia

3.Inclusion or exclusion of maternal blood disorder (i.e.thalassemia).

Response

Yes,

We agreed that these points (the timing of blood test (Hb assessment, definition of anemia ) are varied. We looked for the effects of these factors in meta-regression and they have no effects on the results. Please check the result section table2 page 11. These factors and their effects have been mentioned in the limitations. Please see limitations at page 13 line#235-236.

Comment

  1. The authors did not consider the etiology of maternal anemia.

Response

Yes, we agreed. As you know our objective is the effect of anemia (namely CD) rather than  the determinants of anemia. Perhaps if we looked for etiology  of anemia we could failed to  detect it in  the majority of these papers as authors themselves aimed to  detect the effect of anemia on  CD rather than the etiology of anemia.

Comment

  1. The authors included planned cesarean delivery such as repeat cesarean delivery.

Response

Yes, authors of these papers did include planned cesarean delivery such as repeat cesarean delivery.

Comment

  1. Overall, due to the severe heterogeneity among the studies the effect of anemia on the rate of cesarean delivery is difficult to assess.

Response

Yes, we agreed and we mentioned this as limitation of the study.line#241-241.

Reviewer 2 Report

Well written article 

This rewiev will be beneficial for the literature and clinicians 

This paper investigates the association between maternal anemia and CD. I think it is relevant, original and interesting paper for clinicians. Published articles until October, 2022 are present and analysed for this rewiev. The paper is well written, clear and easy to read. The conclusions are consistent with the evidence and arguments. Statistical analysis is well qualified. This systematic review and meta-analysis shows that anemic pregnant women are more likely to have CD compared with non-anemic pregnant women, appropriately with the current literature. 

Author Response

Reviewer 2

Comment

  1. This paper investigates the association between maternal anemia and CD. I think it is relevant, original and interesting paper for clinicians. Published articles until October, 2022 are present and analysed for this review. The paper is well written, clear and easy to read. The conclusions are consistent with the evidence and arguments. Statistical analysis is well qualified. This systematic review and meta-analysis shows that anemic pregnant women are more likely to have CD compared with non-anemic pregnant women, appropriately with the current literature.

Response

The authors much appreciated your positive feedback. No comments to respond to.

Reviewer 3 Report

I read with huge interest the systematic review and meta-analysis written by Adam et al., concerning a subject that is always interesting and common in current clinical practice. 

From my point of view, the work is fine and the methodology is correct, although I would have made some changes on the study screening in order to improve its extension.

The main criticism goes for the discussion which, in my opinion, must be more developed. I will give further detail in the point-by-point description below. Also, English should be reviewed, as some sentences might be rephrased in order to improve understanding (for example, the first paragraph of the introduction, there are two consecutive sentences that start with "It has been estimated...").

Specific comments:

Abstract:

- The second line (To conduct...) has no conjugated verb. I guess it started with "Objective:" and after removing the subtitle it was left like that. Please fix it.

- Regarding this objective, it is not clear if you look for an association between anemia and CD (being anemia a risk factor for CD), or otherwise. It is clear after reading the paper, but on the objective, I believe that it should be clearer.

- On the methods, the type of articles that have been selected for the systematic review should be made explicit.

Introduction:

- As above-mentioned, I think that English should be revised.

Methods:

- On the study design you indicate that you have selected case-controls studies? Does this mean that you included all studies with a control group? Meaning that cohort studies were also included? I tend to believe so, but if this is the case, it should be better explained. Case-control studies are a type of study, and cohorts one are different. Therefore, the type of study should be better explained.

- Why only studies in English were included? If you want to focus on economic disparities among countries, it would have been more representative if you did not exclude papers in other languages. Nowadays, it is not difficult to have automatic translating services in order to adequate evaluate if a paper is suitable for a systematic review.

- On the statistical methods, you state that haemoglobin cutoff point was 10g/dL, but some papers used 11. Why did you choose to use 10, if the WHO indicates that anemia has to be considered as an Hb level below 11 (you said it yourself in the introduction)?

Results:

- Figure 1: Something happened with paper from the penultimate (n=15) to the last box (n=14). I was not able to find in the text that last step

- Section 3.3, line 4: please give %, not only proportions. It is easier to understand.

Discussion:

In my opinion, this section needs to improve in order to accept the paper. Apart from improving English, the results have to be put in the context of other literature, and the authors have to develop, at least, their thoughts on the association that they have just demonstrated.

Specifically on the limitations, apart from the ones you mentioned, it has to be mentioned that the studies defined anemia in different moments of pregnancy (maybe the impact if different if anemia was at the first trimester or at the third one). Have the authors considered this end and how do they think this may have had an impact on your results?

Author Response

Reviewer 3

Comment

  1. The second line (To conduct...) has no conjugated verb. I guess it started with "Objective:" and after removing the subtitle it was left like that. Please fix it.

 Response

Firstly, the authors much appreciated your positive and helpful feedback.  Yes, we agreed and that is the cause. Now we fixed it as follows: " We conducted this study to explore the possible association between anemia and CD". Please check the abstract section second line.

Comment

  1. Regarding this objective, it is not clear if you look for an association between anemia and CD (being anemia a risk factor for CD), or otherwise. It is clear after reading the paper, but on the
    objective, I believe that it should be clearer.

Response

Yes, we agreed and wrote this objective like this " We conducted this study to explore the possible association between anemia and CD". Please check the abstract second line.

Comment

  1. On the methods, the type of articles that have been selected for the systematic review should be made explicit.

Response

Yes, we agree and wrote it clearer as follows " We included studies with either cross-sectional or case –control or cohort designs.

Comment

  1. Introduction:As above-mentioned, I think that English should be revised.

Response

Yes, we agreed and the whole manuscript is edited by English language native speaker.

Comment

  1. On the study design you indicate that you have selected casecontrols studies? Does this mean that you included all studies with a control group? Meaning that cohort studies were also included? Itend to believe so, but if this is the case, it should be better explained. Case-control studies are a type of study, and cohorts one are different. Therefore, the type of study should be better explained

Response

Yes, we agreed and we included case-conrol studies, cross-sectional studies and cohort study designs. This is wrote clearer in the methods section as requested "" We included studies with either cross-sectional or case –control or cohort designs. Please check the methods section line#70 .

Comment

  1. Why only studies in English were included? If you want to focus on economic disparities among countries, it would have been more representative if you did not exclude papers in other languages.
    Nowadays, it is not difficult to have automatic translating services in order to adequate evaluate if a paper is suitable for a systematic

Response

Yes, we agreed but it might be difficult for us to search for such studies.

Comment

  1. On the statistical methods, you state that haemoglobin cutoff point was 10g/dL, but some papers used 11. Why did you choose to use 10, if the WHO indicates that anemia has to be considered as an Hb level below 11 (you said it yourself in the introduction)?

Response

Yes, we agreed and included the definition of anemia according to CDC and ACOG who used 10.5 levels accordingly we correct this throughout our manuscript. Please check the introduction section line#33-37, methods section line #67-68, statistical analysis line#113-114.

Comment

  1. Figure 1: Something happened with paper from the penultimate (n=15) to the last box (n=14). I was not able to find in the text that last step

 Response

Yes, we agreed and not to confuse the reader we correct the figure 1. Please check figure 1.

Comment

  1. Section 3.3, line 4: please give %, not only proportions. It is easier to understand.

 Response

Yes, we agreed and added the percentages to the ratio. Please check the result section page# 6 line# 157.

Comment

  1. In my opinion, this section needs to improve in order to accept the paper. Apart from improving English, the results have to be put in the context of other literature, and the authors have to develop, at least, their thoughts on the association that they have just demonstrated.

Response

Yes, we agreed and the whole manuscript was edited by a native English speaker. In addition, we add more discussion with regard to our findings. Please check discussion section line#220-232.

Comment

  1. Specifically on the limitations, apart from the ones you mentioned, it has to be mentioned that the studies defined anemia in different moments of pregnancy (maybe the impact if different if anemia was at the first trimester or at the third one). Have the authors considered this end and how do they think this may have had an impact on your results?

 Response

Yes, we agreed and we added a discussion pargrph regarding the specific occurrence of anemia during the course of pregnancy. In addition we add variations in Hb measurement in the included studies as one of the limitations. Please check discssion section page 13 line# 220-232 and line#235-237.  

Round 2

Reviewer 1 Report

The authors improved the manuscript according to the reviewers' suggestions.